# Hedonic processing in humans is mediated by an opioidergic mechanism in a mesocorticolimbic system

Christian Buchel*, Stephan Miedl[†], Christian Sprenger

Department of Systems Neuroscience, University Medical Center Hamburg-Eppendorf, Hamburg, Germany

**Abstract** It has been hypothesized that the pleasure of a reward in humans is mediated by an opioidergic system involving the hypothalamus, nucleus accumbens and the amygdala. Importantly, enjoying the pleasure of a reward is distinct from incentive salience induced by cues predicting the reward. We investigated this issue using a within subject, pharmacological challenge design with the opioid receptor antagonist naloxone and fMRI. Our data show that blocking opioid receptors reduced pleasure associated with viewing erotic pictures more than viewing symbols of reward such as money. This was paralleled by a reduction of activation in the ventral striatum, lateral orbitofrontal cortex, amygdala, hypothalamus and medial prefrontal cortex. Crucially, the naloxone induced activation decrease was observed at reward delivery, but not during reward anticipation, indicating that blocking opioid receptors decreases the pleasure of rewards in humans.

**Editorial note:** This article has been through an editorial process in which the authors decide how to respond to the issues raised during peer review. The Reviewing Editor's assessment is that all the issues have been addressed (see decision letter).

DOI: https://doi.org/10.7554/eLife.39648.001

*For correspondence:
buechel@uke.de

Present address: [†]Department of Clinical Psychology, Psychotherapy and Health Psychology, University of Salzburg, Salzburg, Austria

## Introduction

Goal directed behaviors such as finding a sexual mate or searching for food are crucial for survival and subserved by a mainly dopaminergic motivational system including the ventral striatum (*Morton et al., 2006*). However, in the rodent it has been shown that the ensuing pleasure of these goal directed behaviors that is the hedonic aspect of reward consumption is mediated by an additional, opioidergic system (*Peciña and Berridge, 2005*). This is in line with the observation that opioids play an important role in reproduction, an effect mainly mediated by the hypothalamus and amygdala (*Kostarczyk, 1986*; *Le Merrer et al., 2009*).

In humans, behavioral studies employing opioid antagonists suggest a role of endogenous opioids in relationship to positively valenced stimuli such as attractive faces (*Chelnokova et al., 2014*), food (*Yeomans and Gray, 2002*) and social aspects (*Hsu et al., 2013*). However, evidence with respect to the role of opioids in reward processing in humans is contradictory, as some studies show negative effects of opioid blockade on reward processing (*Petrovic et al., 2008*), whereas others have shown positive effects (*Porchet et al., 2013*).

Additionally, human neuroimaging studies using positron emission tomography with an opioid tracer have revealed a correlation between social acceptance (*Hsu et al., 2013*), positive emotions induced by erotic stimuli (*Koepp et al., 2009*) or food (*Nummenmaa et al., 2018*) and activity in the ventral striatum and the amygdala. However, PET studies have not been able to dissociate the temporal aspects of reward processing such as the effects during anticipation reflecting incentive salience and the hedonic aspects or pleasure related to a reward. This dissociation is neurobiologically important, as previous animal studies have revealed a temporally distinct response pattern of

the mesolimbic system in particular dissociating responses for reward anticipation and outcome (*Schultz et al., 1997*; *Smith et al., 2011*). In this context, it has been shown that anticipatory or conditioned stimuli that predict a reward are associated with incentive salience, but only the actual reward delivery (sucrose) was associated with hedonic effects (*Peciña and Berridge, 2005*; *Smith et al., 2011*). More importantly, the hedonic effect linked to the outcome phase was exclusively opioid dependent, whereas the effect of incentive salience during anticipation was also dopamine dependent.

In humans, anticipation and outcome related neuronal effects can reliably be dissociated by fMRI (*Knutson et al., 2000*; *Yacubian et al., 2006*). Consequently, functional MRI in combination with an opioid antagonist could offer a comprehensive view on the mechanisms of hedonic processing. We therefore performed a within subject, cross-over, pharmacological challenge study with the opioid receptor antagonist naloxone in combination with fMRI and investigated the role of the opioidergic system in processing the pleasure of rewards. To even further dissociate incentive salience from reward related pleasure, we compared erotic stimuli, which directly resemble a pleasurable reward to pictures of monetary outcomes, which are merely visual representations of what a participant can redeem after the experiment (*Sescousse et al., 2010*).

With respect to the task, we aimed for high motivational involvement of the volunteer, which is ideally met by an incentive delay task (*Knutson et al., 2000*). Consequently, we extended the well-established monetary incentive delay task (*Knutson et al., 2000*) with erotic pictures, which have been shown to reliably activate the mesolimbic system (*Redouté et al., 2000*; *Beauregard et al., 2001*; *Arnow et al., 2002*; *Hamann et al., 2004*; *Ponseti et al., 2006*; *Paul et al., 2008*; *Sescousse et al., 2010*; *Sescousse et al., 2013*). In this task, volunteers were cued with the expectable reward magnitude at the beginning of each trial, characterizing the possible outcome. Erotic pictures evoking low pleasure showed women in swimsuits, whereas highly pleasurable stimuli depicted total nudity (*Figure 1*). In monetary trials small and large amounts of money served as rewards. As in classical incentive delay tasks, volunteers had to press a button as soon as a neutral target stimulus appeared on the screen (*Figure 1*). If their response was registered within a defined response window the trial was considered successful and volunteers were shown the reward (erotic picture or picture showing money). Afterwards, they rated the pleasure of viewing the picture in case of reward trials, or how frustrated they were not to be shown the picture in missed reward trials.

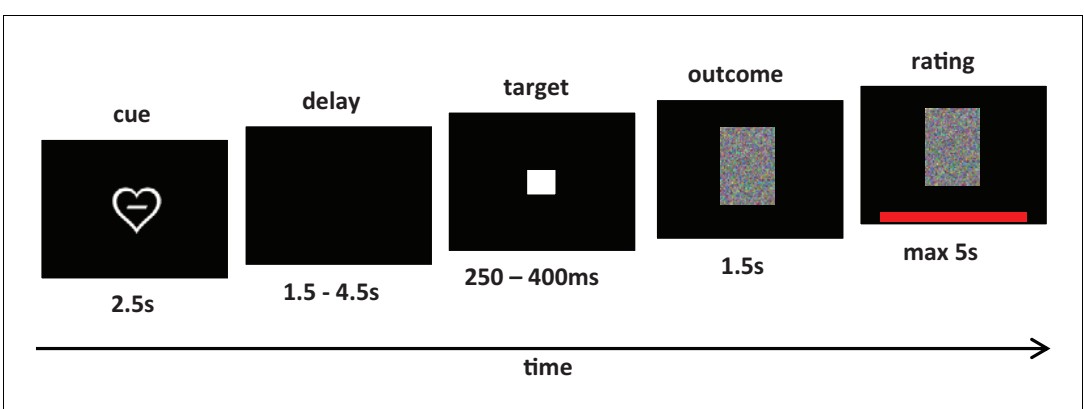

**Figure 1.** Adapted Monetary Incentive Delay (MID) task trial structure. An initial cue signaled potential gain for each trial (high/low pleasure erotic picture or high/low monetary reward). After a variable delay, a target briefly appeared. Responding during target display yielded the indicated gain, whereas late or early responses yielded no gain. Target durations were adapted to approximate 67% hit rate for each subject. In case of gain trials volunteers could watch the outcome picture (erotic picture or money) for 1.5 s. In case of loss trials a scrambled image was shown.

DOI: https://doi.org/10.7554/eLife.39648.002

The following figure supplement is available for figure 1:

**Figure supplement 1.** Behavioral rating data.
DOI: https://doi.org/10.7554/eLife.39648.003

Twenty-one heterosexual, healthy, male volunteers (mean age 25.5 years) took part in the experiment. In monetary trials the amounts of possible monetary rewards were approximately matched for value with the high and low erotic stimuli using a pre-experiment procedure. The order of monetary and erotic trials was randomized and had the same structure, involving high or low possible gains. Volunteers were investigated on two days with either placebo or naloxone. The treatment order was randomized across volunteers. Each erotic stimulus was only presented once and randomized across the placebo and naloxone days.

## Results

### Behavioral and autonomic data

Behavioral mood and side effects ratings did not differ between treatments (*Supplementary file 1 & 2*). Hedonic ratings in the placebo condition indicated that volunteers perceived high erotic stimuli (total nudity) as more pleasurable compared to low erotic (swimsuit) stimuli (mean high erotic >low erotic: T(18)=7.74; p<0.00001). A similar pattern emerged for high and low monetary rewards (*Supplementary file 3*). Not receiving an erotic picture reward led to a comparable pattern of frustration ratings (*Supplementary file 3*). However, absolute ratings for monetary rewards were higher as compared to erotic rewards (*Figure 1—figure supplement 1*).

Interestingly, skin conductance data showed the opposite pattern, with significantly higher values for erotic rewards as compared to monetary rewards (*Figure 2*). This discrepancy was not unexpected as volunteers' ratings on sensitive items such as erotic pictures have been shown to be influenced by social desirability (*Tourangeau and Yan, 2007*). In particular, a high erotic reward induced a significantly stronger SCR response as compared to a high monetary reward (T(13)= 2.87; p=0.007; *Figure 2*). The same was observed comparing low erotic to low monetary rewards (T(13)=

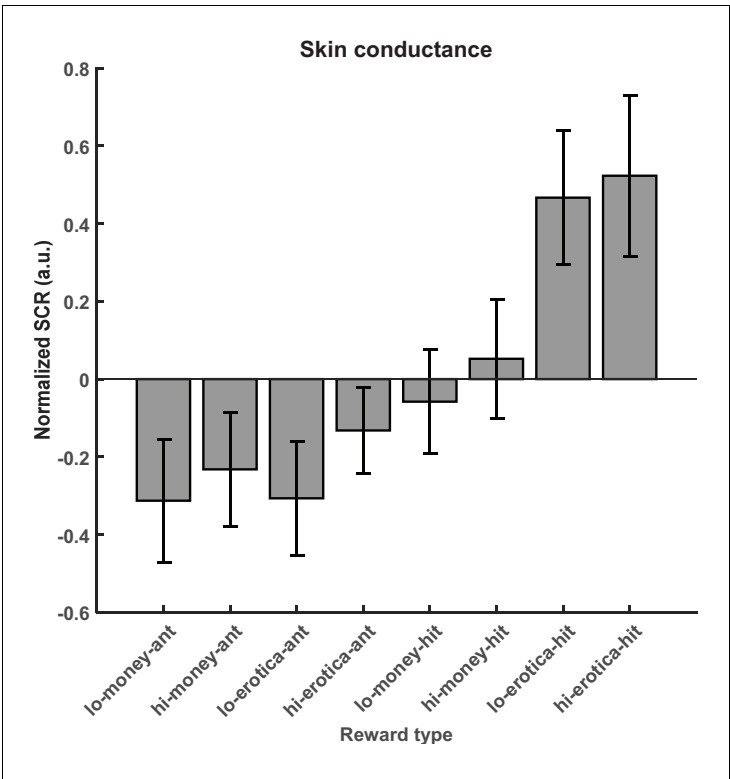

**Figure 2.** Z-transformed skin conductance responses for anticipation and reward. Data is shown for n = 14 volunteers during the placebo treatment only, because naloxone is known to directly affect autonomic regulation of SCR (*Traore et al., 1998*).
DOI: https://doi.org/10.7554/eLife.39648.004

3.16; p=0.004). In addition, only weak responses were observed for the anticipation phase (i.e. evoked by the cue).

In a next step, we investigated the effect of naloxone on perceived pleasure. A repeated measures ANOVA with factors treatment and condition (*Supplementary file 4*) revealed a main effect of condition $F(3.94, 67.12) = 16.709$; $p=1.79*10^{-09}$, a trend for a main effect of treatment $F(1,17)=4.06$; $p=0.06$ and importantly a treatment by condition interaction $F(5.22, 88.68)=2.60$; $p=0.03$. An additional between subject effect of treatment order (i.e. Nlx – Placebo or Placebo - Nlx), revealed neither a significant main effect ($F(1,17)=2.78$; $p=0.11$), nor an interaction with (i) the main effect of treatment ($F(1,17)=1.47$; $p=0.24$), (ii) condition ($F(3.95,67.12) = 1.68$; $p=0.17$) or (iii) the treatment by condition interaction ($F(5.22, 88.68)=1.64$; $p=0.16$).

In particular, pleasure ratings at the outcome phase of the experiment, that is directly after viewing the erotic picture, in gain trials were reduced during naloxone treatment. This effect was most pronounced for the high pleasure condition ($T(18)=3.90$; $p<0.001$; *Figure 3*; *Figure 1—figure supplement 1* and *Supplementary file 5*). Motivation to attain a goal has been shown to affect the level of frustration when thwarted (*Dollard et al., 1939*). We therefore also investigated whether naloxone has an effect on frustration ratings, that is when in missed reward trials the erotic picture was not shown, and observed a significant reduction by naloxone ($T(18)=2.80$; $p<0.006$). In addition we observed a significantly stronger reduction of frustration ratings for the missed reward trials for erotic as compared to monetary outcomes ($T(18)=2.61$; $p=0.0088$; *Figure 3* right; *Supplementary file 5*). A trend for this interaction was also observed for ratings of received rewards ($T(18)=1.66$; $p=0.0572$; *Figure 3* second from right; *Supplementary file 5*). The effect of naloxone on the low erotic stimuli, and monetary rewards was weaker and only significant at the uncorrected level for high monetary rewards ($T(18)=2.14$; $p=0.0229$; *Supplementary file 5*). For low rewards no significant interaction between erotic and monetary outcomes was observed.

## Functional neuroimaging

Based on previous reports (*Redouté et al., 2000*; *Beauregard et al., 2001*; *Arnow et al., 2002*; *Sescousse et al., 2010*; *Sescousse et al., 2013*; *Morelli et al., 2015*; *Noori et al., 2016*) on activations related to erotic and monetary rewards, we focused our analysis on the ventral striatum, the orbitofrontal cortex, the amygdala, the hypothalamus and the medial prefrontal cortex (see Materials and methods and *Supplementary file 6* for the exact definition of these regions of interest based on multiple individual studies and meta-analyses).

Our analysis revealed a reduction of BOLD responses by naloxone to erotic image presentation in bilateral ventral striatum at the outcome phase (right: $T(18)=2.99$; $p=0.004$; $p(corrected)=0.031$; left: $T(18)=3.47$; $p=0.001$; $p(corrected)=0.011$; *Figure 4* and *Figure 4—figure supplement 1*), lateral OFC (right: $T(18)=2.84$; $p=0.006$; $p(corrected)=0.044$; left: $T(18)=1.94$; $p=0.034$; $p(corrected)=0.275$), bilateral amygdalae (right: $T(18)=2.85$; $p=0.005$; $p(corrected)=0.043$; left: $T(18)=2.83$;

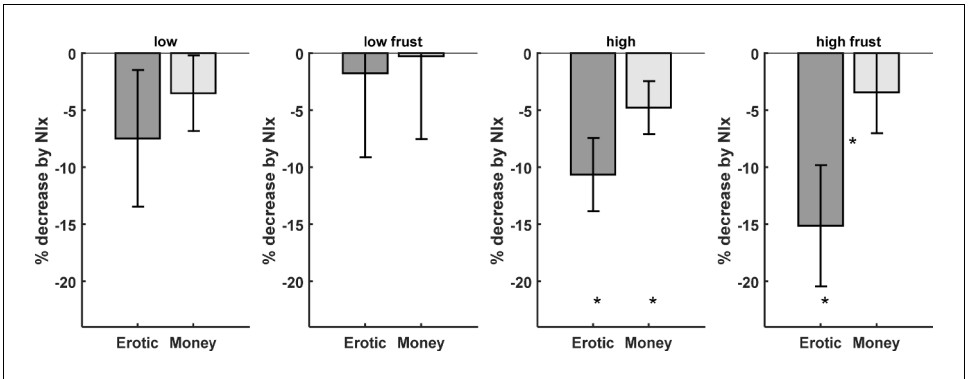

**Figure 3.** Behavioral data shows that the opioid antagonist naloxone significantly reduced ratings for high erotic rewards (dark gray bars). Frustration ratings for missed erotic rewards show a stronger decrease by naloxone as compared to monetary rewards (right). (* denotes p<0.05; * between bars denote p<0.05 for the interaction).
DOI: https://doi.org/10.7554/eLife.39648.005

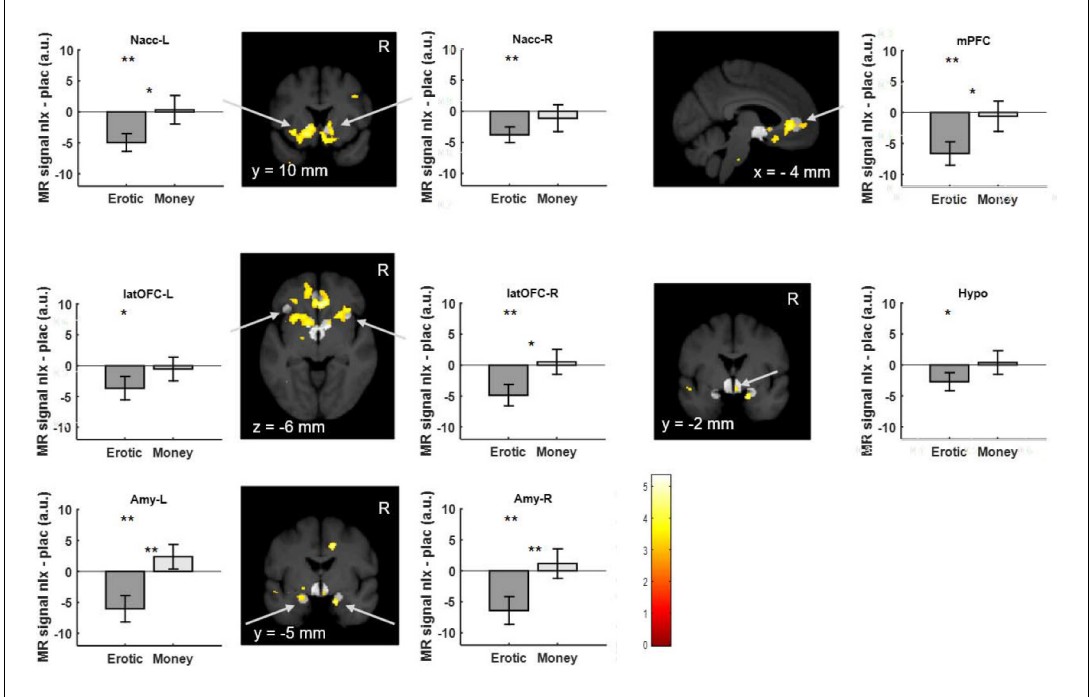

**Figure 4.** Region of interest brain activity for high versus low erotic picture (dark gray) and high versus low monetary reward (light gray) outcomes shows naloxone related reduced activation for erotic trials. Under naloxone treatment, activation in the ventral striatum, medial prefrontal cortex lateral orbitofrontal cortex, the amygdala and the hypothalamus is significantly reduced. This reduction is larger for erotic rewards as compared to monetary rewards. Activations for high versus low erotic pictures comparing placebo to naloxone at p<0.005 (uncorrected, t-test) are overlaid on a mean structural image also indicating the predefined volumes of interest (light gray) in the ventral striatum, medial prefrontal cortex and lateral orbitofrontal cortex, amygdala and hypothalamus. (** denotes p<0.05 corrected for multiple comparisons; *p<0.05 uncorrected; * or ** between bars denote p values for the interaction).

DOI: https://doi.org/10.7554/eLife.39648.006

The following figure supplements are available for figure 4:

**Figure supplement 1.** Region of interest contrasts of brain activity for high versus low erotic picture reward outcome for placebo and naloxone treatment.
DOI: https://doi.org/10.7554/eLife.39648.007

**Figure supplement 2.** Region of interest contrasts of brain activity for high versus low monetary reward outcome for placebo and naloxone treatment.
DOI: https://doi.org/10.7554/eLife.39648.008

**Figure supplement 3.** Region of interest contrasts of brain activity for high versus low monetary reward anticipation for placebo and naloxone treatment (*p<0.05, uncorrected).
DOI: https://doi.org/10.7554/eLife.39648.009

**Figure supplement 4.** Region of interest contrasts of brain activity for high versus low erotic picture reward anticipation for placebo and naloxone treatment.
DOI: https://doi.org/10.7554/eLife.39648.010

**Figure supplement 5.** Right ventral striatum (nucleus accumbens) fMRI responses (arbitrary units) for all stimuli (money/erotic, high/low) and time-points (anticipation, outcome: reward, outcome: miss) under placebo (Plac) and naloxone (Nlx).
DOI: https://doi.org/10.7554/eLife.39648.011

**Figure supplement 6.** Left ventral striatum (nucleus accumbens) fMRI responses (arbitrary units) for all stimuli (money/erotic, high/low) and time-points (anticipation, outcome: reward, outcome: miss) under placebo (Plac) and naloxone (Nlx).
DOI: https://doi.org/10.7554/eLife.39648.012

**Figure supplement 7.** Right lateral orbitofrontal cortex fMRI responses (arbitrary units) for all stimuli (money/erotic, high/low) and time-points (anticipation, outcome: reward, outcome: miss) under placebo (Plac) and naloxone (Nlx).
DOI: https://doi.org/10.7554/eLife.39648.013

**Figure supplement 8.** Left lateral orbitofrontal cortex fMRI responses (arbitrary units) for all stimuli (money/erotic, high/low) and time-points (anticipation, outcome: reward, outcome: miss) under placebo (Plac) and naloxone (Nlx).
DOI: https://doi.org/10.7554/eLife.39648.014

**Figure supplement 9.** Right amygdala fMRI responses (arbitrary units) for all stimuli (money/erotic, high/low) and time-points (anticipation, outcome: reward, outcome: miss) under placebo (Plac) and naloxone (Nlx).

*Figure 4 continued on next page*

*Figure 4 continued*

DOI: https://doi.org/10.7554/eLife.39648.015

**Figure supplement 10.** Left amygdala fMRI responses (arbitrary units) for all stimuli (money/erotic, high/low) and time-points (anticipation, outcome: reward, outcome: miss) under placebo (Plac) and naloxone (Nlx).

DOI: https://doi.org/10.7554/eLife.39648.016

**Figure supplement 11.** Medial prefrontal cortex fMRI responses (arbitrary units) for all stimuli (money/erotic, high/low) and time-points (anticipation, outcome: reward, outcome: miss) under placebo (Plac) and naloxone (Nlx).

DOI: https://doi.org/10.7554/eLife.39648.017

**Figure supplement 12.** Hypothalamus fMRI responses (arbitrary units) for all stimuli (money/erotic, high/low) and time-points (anticipation, outcome: reward, outcome: miss) under placebo (Plac) and naloxone (Nlx).

DOI: https://doi.org/10.7554/eLife.39648.018

p=0.006; p(corrected)=0.044), medial prefrontal cortex (T(18)=3.52; p=0.001; p(corrected)=0.010) and the hypothalamus (T(18)=1.87; p=0.039; p(corrected)=0.309). See *Supplementary file 7* for an overview. For the monetary trials no significant effects of naloxone were observed at the outcome phase (*Supplementary file 9* and *Figure 4—figure supplement 2*). Finally, we investigated whether the opioid antagonist has the same effect on activation during the anticipation phase. This analysis revealed only a weak effect of naloxone in the medial prefrontal cortex (T(18)=1.90; p=0.037; p(corrected)=0.294) and right lateral OFC (T(18)=1.82; p=0.042; p(corrected)=0.339) for monetary trials (*Supplementary file 9* and *Figure 4—figure supplement 3*) but not for erotic trials (*Figure 4—figure supplement 4*). Directly comparing naloxone effects on erotic rewards with its effects on monetary rewards, we observed a stronger effect of naloxone with respect to erotic rewards in all regions of interest (*Figure 4* and *Supplementary file 8*), with the most significant effect in the amygdala (right: T(18)=3.40; p=0.0016; p(corrected)=0.0128; left: T(18)=3.50; p=0.0013; p(corrected)=0.0103). The effects of all conditions in each ROI for saline and naloxone are shown in *Figure 4—figure supplements 5–12*.

In addition we investigated whether the individual decrease of pleasure ratings due to naloxone were related to the reduction of brain activation. Using a linear regression analysis (*Supplementary file 10*), we observed a significant correlation in the hypothalamus (T(17)=3.24; p=0.0024; p(corrected)=0.0194; *Figure 5*; *Supplementary file 10*) and a weaker correlation in the left ventral striatum (T(17)=2.09; p=0.0262; p(corrected)=0.2097; *Figure 5*; *Supplementary file 10*). This indicates that the BOLD signal difference in the hypothalamus due to naloxone is linearly related to the individual decrease in pleasure ratings.

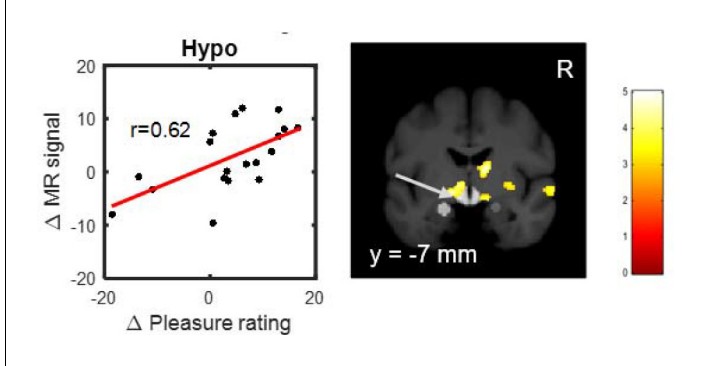

**Figure 5.** Positive correlation between naloxone induced decreases in fMRI signal in the hypothalamus and decreases in ratings for erotic rewards. The more naloxone reduced fMRI signal comparing high versus low erotic rewards during outcome, the more the rating difference between high and low erotic rewards was reduced (See also Table S10). Activations showing this correlation at p<0.005 (uncorrected, t-test) are overlaid on a mean structural image also indicating the predefined volumes of interest (light gray) in the hypothalamus and amygdala.
DOI: https://doi.org/10.7554/eLife.39648.019

# Discussion

Observing that naloxone can reduce pleasure ratings in a cross-over pharmacological challenge design causally implicates the opioidergic system in hedonic processing of erotic stimuli. Furthermore, functional neuroimaging revealed that an opioid receptor antagonist led to a reduction of neuronal activation in the ventral striatum, amygdala, orbitofrontal and medial prefrontal cortex. In addition, the decrease of activation in the hypothalamus due to naloxone was correlated to individual reductions in pleasure ratings, implicating this structure in opioid mediated hedonic processing. Our findings show that a similar system mediates pleasure as it does in rodents (*Peciña and Berridge, 2005*; *Ismail et al., 2009*; *Le Merrer et al., 2009*; *Smith et al., 2011*). Not only did we observe a strong effect of opioid antagonists on pleasurable erotic stimuli, but this effect was specifically related to the outcome phase.

Elegant previous molecular imaging (PET) studies have indicated a relationship between social rejection (*Hsu et al., 2013*), food stimuli (*Nummenmaa et al., 2018*) or erotic stimulus material (*Koepp et al., 2009*) and the opioid system. However, by the nature of the imaging modality, these studies were not in a position to answer the crucial question of whether reward anticipation or outcome is responsible for the observed effect. The rather high temporal resolution of our pharmacological fMRI approach allowed us to dissociate the anticipation from the outcome phase and thus to attribute hedonic reward processing to an opioidergic system.

In a sequential conditioning task in rodents in which a first conditioned stimulus (CS1) was followed by a second (CS2) which was then followed by a sucrose reward, it could be demonstrated that hedonic effects were only observed for the outcome phase, but neither for CS1 nor CS2 (*Smith et al., 2011*). This study has also shown that incentive salience attribution was higher for cues more proximal to the actual reward (i.e. CS2) and although responses to the CS increased after opioid stimulation of the Nacc, they remained much smaller as compared to sucrose reward. Furthermore, the effect of incentive salience at CS2 was dopamine and opioid dependent, whereas the hedonic effect linked to the outcome phase was only opioid dependent. Therefore, one might argue that the effect of opioid blockade at the outcome stage observed in our study could be related to incentive salience (i.e. comparable to conditioned stimulus, CS2) rather than to the hedonic outcome. However, if the erotic reward in our study only represents a CS (e.g. predicting copulatory action) rather than a pleasurable reward per se, this contingency (and any possible incentive salience) would quickly extinguish, because this CS is never reinforced. In addition, attributing incentive salience rather than hedonic outcomes to our erotic rewards is unlikely for other reasons: (i) It has been argued (*Sescousse et al., 2010*) that an important property of erotic pictures is that they directly resemble a reward, which is in contrast to pictures of monetary outcomes that are merely visual representations of what the participant can redeem after the experiment. (ii) Smith and colleagues (*Smith et al., 2011*) observed weak to non-existing facial responses indicative of hedonic processing for CS1 and CS2, but strong responses for the sucrose reward. This closely resembles the pattern of our skin conductance data (*Figure 2*), where very weak responses were observed during the anticipation phase (which could be considered a CS), but very strong responses to the outcome phase (*Figure 2*).

With respect to opposite sex human stimuli, a design involving attractive faces offered first evidence that pharmacological manipulation of the human opioid system can affect motivation for viewing opposite-sex faces (*Chelnokova et al., 2014*). In contrast to these findings, a study involving affective pictures, which included erotic material, could not reveal an effect of naloxone on pleasure ratings (*Kut et al., 2011*). However, the latter study only involved passive viewing of affective pictures, which is in contrast to our study, where volunteers had to successfully perform a task to view erotic pictures. This suggests that the role of opioids in mediating pleasure is modulated by motivational aspects, analogous to observations in the dopaminergic system (*Coricelli et al., 2005*).

Previous studies observed a correlation of hypothalamic activity with pleasure ratings and penile tumescence in the context of visual erotic stimuli (*Redouté et al., 2000*; *Arnow et al., 2002*; *Paul et al., 2008*; *Georgiadis et al., 2012*) and direct sexual activity (*Georgiadis et al., 2010*). Importantly, our data suggests that this effect is mediated by opioids, because the individual reduction of pleasure ratings for erotic stimuli by naloxone was correlated with the reduction of brain activation in the hypothalamus.

The hypothalamus is an important structure for behavioral, autonomic and endocrine responses in relation to reproductive behavior (*Le Merrer et al., 2009*). Opioid receptor agonists injected directly into the hypothalamus inhibited or delayed sexual behavior (*van Furth et al., 1995*; *Le Merrer et al., 2009*). In contrast, opioid antagonists facilitate male sexual behavior. In addition, rodent studies have revealed that opioid receptor antagonists (*Agmo and Gómez, 1993*) but not dopamine antagonists (*Ismail et al., 2009*) in the hypothalamus can block conditioned place preference (CPP) linked to sexual behavior, indicating a distributed system of hypothalamic and limbic regions for mediating the effects of sexual rewards (*Le Merrer et al., 2009*). Given that sexual behavior in rodents is increased by opioid antagonists, whereas we observed decreased pleasure in viewing of erotic pictures under naloxone emphasizes that additional processes contribute to sexual behavior as compared to viewing erotic pictures. The pivotal role of the hypothalamus is further underlined by the observation that infusion of an opioid antagonist into the nucleus accumbens did not reduce the reinforcing properties of ejaculation. This strong observation led the authors to conclude that the hypothalamus is the site where sexual reward is produced (*Agmo and Gómez, 1993*). This resonates with our finding that the hypothalamus was the region where we observed the strongest individual relationship between reduction of pleasure ratings by naloxone and the ensuing reduction of the BOLD signal.

We investigated fMRI activation differences between high and low magnitude trials. The alternative, that is comparing high (or low) magnitude trials to a resting baseline includes unspecific effects such as those evoked by gross differences in visual stimulation. Therefore, showing that an opioid antagonist can block differential responses for high versus low erotic stimuli directly implies opioids in mediating these effects. Furthermore, our behavioral data is in line with previous studies, showing that the effect of an opioid antagonist was strongest for the most valuable stimuli (*Chelnokova et al., 2014*).

Interestingly, we observed higher average ratings for monetary rewards as compared to erotic rewards, which seems to contradict our attempt to equate the value of monetary and erotic stimuli in a pre-experiment. However, previous studies have also observed a dissociation of ratings and the amount of work volunteers are willing to perform to increase viewing time of attractive faces (*Aharon et al., 2001*). Nevertheless, our calibration procedure was sufficient to equate the range of ratings for monetary and erotic stimuli to be captured by the same visual analogue scale and thus avoid ceiling or floor effects. Importantly, the focus of our study was on how ratings change under naloxone treatment within a stimulus category and our results clearly indicate that although absolute ratings were higher for monetary rewards, the relative difference due to opioid blockade is significantly larger for erotic rewards. Furthermore, our skin conductance data from the placebo session clearly indicates that SCR responses are significantly larger for erotic rewards as compared to monetary rewards. These findings together with the observation that fMRI responses in all regions of interest were stronger for erotic stimuli indicate that lower ratings for erotic stimuli as compared to monetary stimuli might at least in part be related to social desirability (*Tourangeau and Yan, 2007*). However, we cannot rule out that there is a genuine difference between ratings and autonomic and neural responses with respect to erotic and monetary stimuli.

When comparing high to low magnitude monetary rewards, for the placebo treatment we only observed weak effects. This is not surprising as previous studies have established that brain areas including the ventral striatum and vmPFC adapt their dynamic range to the overall value range of the stimuli employed in an experiment (*Benedek and Kaernbach, 2010*; *Bostwick and Bucci, 2008*; *Boucsein et al., 2012*; *Calhoun et al., 2017*). Based on these observations it is to be expected that highly rewarding and pleasurable outcomes such as erotic pictures can adaptively down-regulate the responses to less arousing monetary rewards. This notion is also supported by the skin conductance data indicating that (i) SCR responses are significantly larger for erotic rewards as compared to monetary rewards and (ii) responses to erotic rewards are much larger compared to activation during anticipation (*Figure 2*).

Although our approach revealed an effect of opioids on processing of pleasurable rewards this does not rule out the role of other modulatory neurotransmitter systems such as the endocannabinoid system which is known to have effects on emotional processing (*Laviolette and Grace, 2006*). A further limitation of our study is the considerably small sample size. This is unfortunately caused by the great effort of a cross-over pharmacological challenge study using fMRI. However, using a longitudinal design we could minimize between subjects variance.

Our data also sheds light on the emerging clinical application of opioid antagonists in treating internet sex and pornography addiction (*Bostwick and Bucci, 2008*; *Raymond et al., 2010*; *Kraus et al., 2015*) as well as in treating adolescent sexual offenders (*Ryback, 2004*). Numerous case reports have documented a very high effectiveness of the opioid antagonist Naltrexone for treating severely affected patients. Importantly, in the course of treatment patients have described a diminished sense of 'overwhelming pleasure' (*Kraus et al., 2015*) which is in line with our data showing a major effect of opioid blockade on the hedonic aspects of reward processing. In the context of a classical conditioning framework the pleasure of reward can be considered as part of the unconditioned stimulus (UCS) whereas cues that predict this represent conditioned stimuli (CS). Consequently, by reducing reward pleasure the predictors of reward will also be devalued which in turn can lead to an overall therapeutic success including a reported reduction of craving (*Kraus et al., 2015*).

## Materials and methods

### Participants

Currently, no study has investigated the effect of naloxone on pleasure ratings related to erotic stimuli or monetary rewards. For our power calculation, we were therefore guided by the effect sizes reported in a previous study on the effect of naloxone on affective ratings in a monetary gambling task (*Petrovic et al., 2008*). By careful visual inspection of *Figure 1* in *Petrovic et al., 2008*, we estimated the mean effect of naloxone on ratings ($\mu$-$\mu_0$) to $-3.53$, with a standard deviation ($\sigma$) of 6.31. Based on the following equation and setting power (1-$\beta$) to 80% and type I error rate ($\alpha$) to 5% we estimated the sample size (n)

$$n = \left( \sigma \cdot \frac{Z_{1-\alpha} + Z_{1-\beta}}{\mu - \mu_0} \right)^2$$

(http://powerandsamplesize.com) to be 19.75.

Consequently, 21 heterosexual male volunteers (Mean ± SD age, 25.48 ± 4.55 years) with no history of neurological or psychiatric disorders participated in this study. Sexual arousability was measured with the Sexual Arousability Inventory (SAI) (*Hoon and Chambless, 1998*) (mean score SAI: 88.45 ± 11.91), ensuring that subjects showed a normal sexual arousability.

Data for two volunteers could not be used due to technical problems (scanner artefacts), leaving a final sample size of nineteen. The study was conducted in accordance with the Declaration of Helsinki. All subjects gave written informed consent to be part of the study, which was approved by the ethics committee of the Chamber of Physicians, Hamburg, Germany (PV3906). The informed consent also included the consent to publish the data.

### Task

Our task extended the classical monetary incentive delay task (MID) (*Knutson et al., 2000*) by a condition in which erotic pictures were employed instead of money. As in the classical MID task, trials started with a cue phase. The cue indicated the nature and magnitude of the possible reward in this trial: A heart indicated a possible erotic stimulus, whereas a Euro symbol (€) indicated a possible monetary reward. Magnitude of either type of stimulus was indicated by either a single (low) or three (high) horizontal lines. In erotic trials low magnitude was defined as showing female models in swimsuits, whereas high magnitude related to showing completely nude pictures. 110 erotic pictures were selected from the Internet according to two objective criteria: women had to be alone and their face was not displayed. After a variable delay between 1.5 and 4.5 s the target (white square) appeared and volunteers had to indicate this with a button press as quickly as possible. If the response occurred within a response window, the trial was considered successful and volunteers were shown the picture. In a pretest we estimated a time windows so that 67% of all responses were valid. In case of an erotic picture trial they were able to view the picture for 1.5 s, in case of a monetary trial they were shown their monetary gain for the same time period. In case of a non-successful trial a scrambled picture was shown. Immediately afterwards, they rated the pleasure of the outcome (i.e. either viewing the erotic picture or the picture of the monetary reward in gain trials, or how

frustrated they felt in missed reward trials). Rating was performed by moving a cursor on a continuous visual analog scale ranging from 0 to 100.

We approximately equated pleasure ratings for the two modalities to be able to use the same visual analogue scale in both conditions (i.e. to guarantee that the scale covers the range of pleasure ratings for both modalities and to avoid any ceiling or floor effects). In a pre-experiment session we estimated how much work (clicks) volunteers are willing to perform (*Aharon et al., 2001*) for fixed monetary values (0.32, 0.39, 0.49, 0.61, 0.76, 0.94, 1.17, 1.46, 1.81, 2.26, 2.81, 3.49, 4.35, 5.41, 6.73, 8.37, 10.41, 12.95, 16.11, 20.04) and erotic pictures (high and low). The logarithmic grading of these fixed values was chosen to be more sensitive in the low value range. In this pretest we used a modified incentive delay task in which volunteers had to press a button as many times as they want in a 5 s period. They were told that the more often they press within a 5 s period, the more likely it will be to gain money or be able to view the erotic picture. We estimated the maximum number of button presses (in a 5 s interval) for each individual before this test by asking them to press a button as often as possible in a 5 s period. In analogy to a Becker-deGroot-Marschak auction (*Becker et al., 1964*) for each trial we generated a random number in the interval between 0 and 150(%) of their maximum button press rate. If that number was lower than the amount of button presses for this trial a gain occurred. Consequently, volunteers would gain in 66% of trials if they performed at their maximum response rate in each trial.

We then fitted an exponential function to the monetary data. Finally, we took the intersection of the amount of work (clicks) volunteers were willing to spend to see the high and low erotic pictures and the fitted exponential function to identify the monetary equivalent of watching a high or a low erotic picture. In case this procedure could not reveal meaningful amounts, default values of 0.4€ for low and 1.8€ for the high monetary amount were selected. The actual amounts used ranged from 0.1 to 5.6€ (0.85 ± 1.09€; mean ± sd) for the low amount and from 0.8 to 8.1€ (mean ± sd 2.71 ± 1.32€) for the high amount.

## Drug administration

Volunteers were investigated in a cross-over design on two days (~48 hr apart) with either the application of naloxone or placebo (order randomized across volunteers). At ~15 min before the start of the experiment, we administered a bolus dose of 0.15 mg/kg naloxone (Naloxon-ratiopharm, Ratiopharm, Ulm, Germany) or the same volume of saline via an intravenous line inserted into the antecubital vein of the left arm. Because naloxone has a relatively short half-life (~70 min in blood plasma; Summary of Product Characteristics, Ratiopharm) and its clinically effective duration of action can be even shorter (*Gutstein and Akil, 2006*), we additionally administered an intravenous infusion dose of 0.2 mg/kg/h naloxone for the duration of the experiment (diluted in saline) or the same volume of saline, starting shortly after bolus administration. This dosing regime leads to a stable concentration of naloxone in blood plasma over the length of the experiment (*Eippert et al., 2009*; *Schoell et al., 2010*) and is sufficient to block central opioid receptors almost completely (*Mayberg and Frost, 1990*).

Subjects were informed about naloxone, including its pharmacological properties, its general clinical use, and its possible side effects. Subjects were also informed that they would most likely not notice that they had received naloxone, as it generally does not have noticeable effects on mood at this dose (*Grevert and Goldstein, 1978*; *Petrovic et al., 2008*; *Kut et al., 2011*). After each experiment the experimenter (S.M.) recorded mood and possible side effects using a 5-point Likert scale (not at all - very) with 12 items for mood and a 7-point Likert scale (not present - extreme) with seven items for side effects. The mood rating scale included the following items: satisfied, rested, restless, bad, worn out, calm, tired, good, uneasy, cheerful, unwell and relaxed. The seven item side effect scale included the following items: dry mouth, dry skin, blurred vision, lethargy, sickness, dizziness, and headache. Naturally, we could not inform subjects about the true purpose of naloxone administration in this study, which was done during debriefing. The experimenter (S.M.) who interacted with the subjects was blinded as to which drug was given. Blinding and assignment of treatment order by a random number was performed by another experimenter (C.S.). Unblinding occurred after the experiment. In the final sample ten volunteers received saline on day 1 and naloxone on day 2, nine volunteers received naloxone on day 1 and saline on day 2.

## Behavioral data

Pleasure and frustration ratings for each trial type (monetary low, monetary high, erotic low, erotic high) for naloxone and placebo were compared using a repeated measures ANOVA with factors trial type and treatment. Given the strong a priori hypothesis that naloxone would decrease pleasure ratings (*Petrovic et al., 2008*), we additionally performed planned individual paired one-sided t-tests. Results were corrected for multiple comparisons using a Bonferroni correction.

## Autonomic data

Electrodermal activity was measured during fMRI with MRI-compatible electrodes on the palm of the left hand (thenar and hypothenar sites) connected to carbon leads (Biopac, Lead108). The signal was amplified using an analog amplifier (Biopac, MP150) and sampled at 1000 Hz using CED 1401 analog-digital converter (Cambridge Electronic Design). After temporal smoothing using a Gaussian convolution kernel with a full-width-at-half-maximum of 0.4 s and subsequent downsampling to 10 Hz, we computed the phasic skin conductance drive (SCR) using a deconvolution technique (*Benedek and Kaernbach, 2010*) as implemented in Ledalab 3.4.8 and used these to assess the autonomic arousal associated with individual stimuli in a time window from 1 to 4 s after stimulus onset (*Boucsein et al., 2012*). To account for differences in electrode position, skin moisture and other between subject effects (*Boucsein et al., 2012*), SCR estimates were z-transformed within volunteers and session, and then averaged across sessions. We only analyzed SCR data from the placebo treatment condition, as it has been shown that naloxone suppresses the descending bulbar inhibitory mechanisms on SCR responses (*Traore et al., 1998*) and thus alters SCR responses. Due to artefacts (i.e. MR gradient switching artefacts, cable movement), skin conductance data was only available from 14 volunteers for the placebo day.

## Imaging

Scanning was performed with a 3T whole-body magnetic resonance imager (TIM Trio, Siemens, Erlangen, Germany). fMRI data acquisition was divided into four sessions. In each session we acquired between 311 and 332 volumes (depending on response timings) per session with 36 slices in descending order (2 mm slice thickness with 1 mm gap) using a gradient-echo T2*-weighted pulse sequence (EPI). The time to repetition (TR) for volume acquisition was set to 2160 ms and the time to echo (TE) to 25 ms. In-plane resolution was $108 \times 108$ with a field of view of $216 \times 216$ mm. For anatomical reference, a 3D magnetization prepared gradient-echo sequence of the whole brain was obtained with TR of 6.8 ms and a TE of 3.2 ms.

Image preprocessing and analyses were performed with SPM12 software (Wellcome Trust Centre for Neuroimaging, London). For structural preprocessing, we used DARTEL to spatially normalize individually segmented T1-weighted scans to a template (Template_X_IXI555_MNI152.nii; http://brain-development.org/) as provided by the CAT 12 toolbox (http://dbm.neuro.uni-jena.de/cat12). Functional images were realigned and resliced in a two pass approach (initially to the first volume, then to the mean of all volumes). In addition, a mean functional image was created for each volunteer and used to derive a deformation field for spatial normalization into MNI space using the unified segmentation approach. A direct estimation of the deformation field from the functional images has been shown to outperform a combined coregistration approach in some cases (*Calhoun et al., 2017*). Single-subject statistical models analyzed the resliced data for anticipation (i.e. when the cue was shown) and outcome (i.e. when the outcome was presented) for low and high monetary and erotic trials. Each condition was defined separately for successful (gain) and unsuccessful (no gain) trials. Trials in which subjects failed to respond were modeled as error trials. Rigid body movement parameters from the realignment procedure were included as six additional nuisance covariates. Next, contrast images of the parameter estimates were created for each subject. Single-subject contrast images were created by applying the deformations as estimated from the unified segmentation normalization of the mean functional images to the contrast images, which were subsequently resampled with a resolution of $1.5 \times 1.5 \times 1.5$ mm$^3$ and smoothed with a Gaussian kernel of 8 mm full-width-at-half-maximum (FWHM). Normalized and smoothed single-subject contrast images were then entered into a second-level random effects analysis (paired t-test contrasting the naloxone with the placebo treatment condition) reflecting a stimulus (high minus low erotic stimuli or high minus low monetary reward) by treatment (naloxone versus placebo) interaction analysis. Furthermore, we

tested for an interaction comparing the naloxone versus placebo contrast image using a paired t-test. Finally, we performed a regression analysis, in which we added a covariate to the second level statistical model, coding the rating difference between the naloxone and the placebo scan.

Functional imaging analyses were based on regions of interest (ROIs) based on averaged coordinates from previous individual studies (*Redouté et al., 2000*; *Beauregard et al., 2001*; *Arnow et al., 2002*; *Sescousse et al., 2010*; *Sescousse et al., 2013*) and meta-analyses (*Kühn and Gallinat, 2012*; *Morelli et al., 2015*; *Noori et al., 2016*) on monetary and erotic rewards. According to these studies spheres of 6 mm radius around peak coordinates were located in ventral striatum (left: −10 10–5; right: 10 7–6 mm), lateral OFC (left: −36 25–3; right: 34 19–5 mm) and amygdala (left: −19–4 −20; right: 21–1 −18 mm). In addition we employed 10 mm radius spherical ROIs centered on the medial prefrontal cortex (1 40 0 mm) and the hypothalamus (0–6 −8 mm) (*Supplementary file 7*).

## Acknowledgements

We would like to thank the radiographer team at the Department for Systems Neuroscience for help with scanning, Brian Knutson and Stefanie Brassen for comments on an earlier version of this manuscript and Christian Gaser for his CAT12 toolbox. CB is supported by the DFG, SFB T-CRC 134 project C08 and SFB 936 project A06. CS was supported by the ERC, ERC-2010-AdG_20100407.

## Additional information

### Competing interests
Christian Buchel: Reviewing editor, *eLife*. The other authors declare that no competing interests exist.

### Funding

| Funder | Grant reference number | Author |
| --- | --- | --- |
| Deutsche Forschungsgemeinschaft | SFB 936 project A6 | Christian Buchel<br>Christian Sprenger |
| H2020 European Research Council | ERC-2010-AdG_20100407 | Christian Buchel<br>Christian Sprenger |
| Deutsche Forschungsgemeinschaft | SFB TR 134 Project C08 | Christian Buchel |

The funders had no role in study design, data collection and interpretation, or the decision to submit the work for publication.

### Author contributions
Christian Buchel, Conceptualization, Supervision, Funding acquisition, Investigation, Visualization, Writing—original draft, Project administration, Writing—review and editing; Stephan Miedl, Data curation, Investigation, Methodology, Writing—review and editing; Christian Sprenger, Supervision, Investigation, Methodology, Writing—original draft, Writing—review and editing

### Author ORCIDs

Christian Buchel http://orcid.org/0000-0003-1965-906X

### Ethics
Human subjects: The study was conducted in accordance with the Declaration of Helsinki. All subjects gave written informed consent to be part of the study, which was approved by the ethics committee of the Chamber of Physicians, Hamburg, Germany (PV3906). The informed consent also included the consent to publish the data.

Decision letter and Author response
Decision letter https://doi.org/10.7554/eLife.39648.034
Author response https://doi.org/10.7554/eLife.39648.035

## Additional files

### Supplementary files

• Supplementary file 1. Side effects ratings. Two sided paired Wilcoxon signed rank test comparing mood between naloxone (_nlx) and saline (_nacl) sessions.
DOI: https://doi.org/10.7554/eLife.39648.020

• Supplementary file 2. Mood ratings. Two sided paired Wilcoxon signed rank test comparing mood between naloxone (_nlx) and saline (_nacl) sessions.
DOI: https://doi.org/10.7554/eLife.39648.021

• Supplementary file 3. Rating differences for successful and missed reward trials in the placebo treatment condition.
DOI: https://doi.org/10.7554/eLife.39648.022

• Supplementary file 4. Effects of Naloxone on pleasure ratings. Repeated measures ANOVA, Huynh-Feldt nonsphericity correction.
DOI: https://doi.org/10.7554/eLife.39648.023

• Supplementary file 5. Naloxone effects (relative to placebo) on pleasure ratings.
DOI: https://doi.org/10.7554/eLife.39648.024

• Supplementary file 6. Coordinates from individual studies to define the center of ROIs.
DOI: https://doi.org/10.7554/eLife.39648.025

• Supplementary file 7. Naloxone effects (relative to placebo) on the high versus low erotic image outcome contrast.
DOI: https://doi.org/10.7554/eLife.39648.026

• Supplementary file 8. Naloxone effects (relative to placebo) comparing erotic versus monetary outcomes.
DOI: https://doi.org/10.7554/eLife.39648.027

• Supplementary file 9. Naloxone effects (relative to placebo) on other contrasts.
DOI: https://doi.org/10.7554/eLife.39648.028

• Supplementary file 10. Regression analysis relating the change in pleasure ratings to the change in fMRI signal.
DOI: https://doi.org/10.7554/eLife.39648.029

• Transparent reporting form
DOI: https://doi.org/10.7554/eLife.39648.031

### Data availability

Only freely available data analysis tools were used (SPM, Ledalab). Data are available online via Dryad (http://dx.doi.org/10.5061/dryad.11j304c)."

The following dataset was generated:

| Author(s) | Year | Dataset title | Dataset URL | Database and Identifier |
|---|---|---|---|---|
| Christian Buchel, Stephan Miedl, Christian Sprenger | 2018 | Data from: Hedonic processing in humans is mediated by an opioidergic mechanism in a mesocorticolimbic system | http://dx.doi.org/10.5061/dryad.11j304c | Dryad Digital Repository, 10.5061/dryad.11j304c |

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
