## [Decision Letter]

[**Editorial note:** This article has been through an editorial process in which the authors decide how to respond to the issues raised during peer review. The Reviewing Editor's assessment is that all the issues have been addressed.]

Thank you for submitting your article "Hedonic processing in humans is mediated by an opioidergic mechanism in a mesocorticolimbic system" for consideration by *eLife*. Your article has been reviewed by three peer reviewers, including Geoffrey Schoenbaum as the Reviewing Editor, and the evaluation has been overseen by a Senior Editor. The reviewers have opted to remain anonymous.

The Reviewing Editor has highlighted the concerns that require revision and/or responses, and we have included the separate reviews below for your consideration. If you have any questions, please do not hesitate to contact us.

Summary:

In this study, the authors test the hypothesis that hedonic responses to rewards are opioid mediated in a circuit including the hypothalamus, accumbent, and amygdala in humans. Human subjects were trained in an "incentive delay task" to associate pictures with high and low value monetary and erotic outcomes. After learning the task, they performed after challenge with naloxone or placebo while BOLD signal, SCR and subjective ratings were acquired. The authors report that naloxone blunted responses, especially to high value erotic outcomes during the reward phase of the task. This was associated with BOLD signal changes in the hypothesized regions.

Major concerns:

In the discussion, there were a number of concerns that were generally about the selective manner in which the results – particularly behavioral – were presented and analyzed. While the SCRs are presented in a relatively clear manner, they are analyzed by t-test. Reward ratings were presented and analyzed pairwise. And there was no distinction between outcome and anticipatory phase data. This then carried on into the manner in which the BOLD was analyzed. One might argue this was targeted, but the targets were not a priori and the justification was not clear. This selective focus made it hard to evaluate the task (reviewer 1, point 1) and caused some questions about whether other contrasts might have been more appropriate or shown different results (reviewer 1, point 1 and reviewer 3, point 2 and 6). It was felt that a fuller analysis and consideration of these questions was necessary. Relatedly, reviewer 2 felt that softening the strong distinction the authors are making between phases that distinguish incentive versus hedonic responses. This would perhaps go along with a fuller presentation and analysis of anticipatory vs outcome related behavior. Finally reviewer 3 had a variety of concerns regarding limitations that should be added to the manuscript – the impact of naloxone on mood, which presumably might underlie what they see and the low N for example should be mentioned.

Separate reviews (please respond to each point):

*Reviewer #1:*

In this study, the authors test in humans the hypothesis that pleasure during outcome presentation is selectively mediated by opioid mechanisms. Subjects performed a simple delay task in which they responded to cues that predicted the delivery of high versus low value erotic versus monetary rewards. After obtaining baseline measures, the same task was done following challenge with naloxone or placebo. They report that naloxone significantly affected ratings and SCR for the erotic but not the monetary outcomes, and this effect was associated with reduced activation in ventral striatum, orbitofrontal cortex, amygdala, hypothalamus and medial prefrontal cortex. The result is generally consistent with the idea that opioids mediate the pleasure of receiving reward distinct from anticipation and incentive salience and provides evidence of this in humans along with the potential circuit mediating the effects. Overall the experiments seem well designed and cleanly conducted and the results important.

However, while I found it interesting and potentially it seems like a novel and important study, I was a bit frustrated with how the results – particularly the behavioral results – were presented. Possibly my concerns are quite minor and may just involve a fuller presentation and analysis of the results. But if I follow the design correctly, it seems to me that the analysis should be a simple multifactor ANOVA comparing effects of placebo and drug on the different measures (SCR, ratings) over the different rewards – money and pictures. Yet instead the presentation of these data focuses in on specific comparisons and reports t-tests. I don't think these were a priori comparisons really, so this is not really appropriate. I think the authors need to present the results from each type of trial and analyze for the effects of the different factors. This could be extended to showing SCR measures during the anticipatory versus outcome periods as well. This level of detail would make the results much easier to appreciate and more meaningful I think. As it is, I feel like I do not have a full picture of the behavior.

The second issue I have is with the explanation for why the ratings and results were so different for the two types of rewards. The authors seem to minimize this (e.g. social desireability – what is this – embarrassment?), but I wonder whether this difference is important and potentially reflects the fact that different systems are mediating the response to one (erotic, primarily emotional and directly experienced) versus more cognitive and abstract to the other (money, the actual outcomes must be thought about). The authors allude to this at one point, but then do not say much about it. If this is a real difference, it should be addressed more, especially as many fMRI studies use money as the primary reward.

Additional data files and statistical comments:

My major concern is partly statistics – as outlined above.

*Reviewer #2:*

This manuscript by Buchel, Miedl, and Sprenger presents an interesting and compelling demonstration that hedonic reactions to pleasant outcomes are opioid mediated in people.

The experiments seem well designed, the data are convincing, and the entire article is clear and scholarly. The authors deserve congratulations on a fine study. I have no major criticisms.

Minor point:

While I agree with the authors' primary conclusion that they demonstrate an opioid mediation of outcome hedonic impact (the naloxone effects are especially compelling), and the authors also are perfectly right to divide anticipation vs. outcome temporal phases into psychological incentive salience vs. hedonic impact predominant categories. However, it should also be noted that the temporal anticipatory/consummatory divide is only a rough guide for motivation/hedonic categories, and temporal moments dos not always map on perfectly to the psychological categories.

That is, even if anticipatory phase is primarily 'wanting', there can be some cue-elicited hedonic impact in the anticipatory phase (albeit low level) as a conditioned hedonic reaction as the authors note. There can also be an augmentation of incentive salience in the outcome phase, as in drug-priming of craving, or the 'you can't eat just one chip' or cocktail peanut priming effect of a food morsel. It is plausible that sexual incentive salience could also be heightened during outcome phase of sexual images. In my view, this consideration might justify slightly softening the authors assignment of psychological components to anticipatory/consummatory phases, but does not detract from their main conclusion.

Similarly, regarding the authors' description in Discussion: "In a sequential conditioning task in rodents in which a first conditioned stimulus (CS1) was followed by a second (CS2) which was then followed by a sucrose reward, it could be demonstrated that hedonic effects were only observed for the outcome phase, but neither for CS1 nor CS2 (Smith et al., 2011)..….Furthermore, the effect of incentive salience at CS2 was dopamine and opioid dependent, whereas the hedonic effect linked to the outcome phase was only opioid dependent."

I think in the Smith et al., 2011 study, there actually was also an opioid enhancement of conditioned anticipatory hedonic reactions to CS1 and CS2 (as well as a larger hedonic enhancement to sucrose outcome). The conditioned hedonic reactions were very low to begin with, as the authors note, but became slightly larger after NAc opioid stimulation. As the authors say, the UCS outcome hedonic enhancement was much greater, in keeping with their conclusions. Possibly the description could be slightly adjusted.

*Reviewer #3:*

Buchel et al. investigated the impacts of the opioid receptor antagonist naloxone on reward processing in healthy human volunteers using an adapted incentive delay task with sexual and monetary stimuli. Naloxone treatment was associated with a stronger reduction of subjectively reported pleasure while viewing erotic pictures compared to viewing symbols of monetary reward. Moreover, naloxone decreased the activation of reward-related areas primarily during "reward delivery".

The manuscript is well-written and the research question is of high interest. The imaging methods are sound. However, the behavioral pharmacology part has some major flaws. Below you will find the several major concerns that have to be addressed.

1) The authors did not assess mood/affective state ratings, which is a major shortcoming of the study. The reviewer disagrees with the statement that "[naloxone] generally does not have noticeable effects on mood (Grevert and Goldstein, 1978; Petrovic et al., 2008; Kut et al., 2011)". There are a number of other studies clearly showing that naloxone infusions can induce aversive states such as dysphoria (Cohen et al., 1983, Arch Gen Psychiatry; Martin del Campo et al., 1994, Psychopharmacology; 2000, J Affect Disorder). Thus, the overall effect of mood on pleasure ratings remains unclear here. This has to be discussed as a strong limitation.

2) The reviewer is concerned regarding the choice of the sexual stimuli and the contrast build on the related picture categories (low vs. high erotic reward). Most likely, the supposed "low erotic reward" (women in swimsuits) category is already inducing strong activations of the reward system. This interpretation is supported by the fact that women in swimsuits ("low erotic reward") induced much stronger SCR responses than high monetary rewards but that there was no difference compared to high erotic reward pictures. Interestingly a recent study has shown that explicit erotic pictures (comparable with "high erotic reward" stimuli used here) did not induce higher sexual arousal ratings in comparison to implicit erotic pictures (comparable with "low erotic reward" stimuli) (Schmid et al., 2015, Eur Neuropsychopharmacol). Thus, the question arises as to whether the chosen contrast is ideal for investigating the rewarding nature of sexual stimuli. More neutral pictures might have been a better choice. This has to be discussed as a major limitation.

3) In light of the replication crisis in human neuroscience, the sample size is considered small (n=19) – a further limitation that has to be discussed. Related to that point, the authors report corrected and uncorrected p-values for the behavioral analyses but tested only one-tailed hypotheses. A more conservative approach would be appreciated, e.g., reporting and discussing only corrected results.

4) Did the authors control for potential effects of order? Specifically, the very short test-retest interval (48h) suggests that order effects should be investigated, as naloxone may impact up- or down-regulation of opioid receptors far beyond 48h (Sirohi et al., 2007, J Pharmacol Exp Ther; Jenab and Inturrisi 1994, Mol Brain Res; Belcheva et al., 1992; Mol Pharmacol). Thus, the administration of naloxone on the first test day may have influenced results from the placebo condition on the second test day. The very short interval is another major weakness of the study.

5) Interpretation and nature of effects: While the behavioral data suggest that naloxone reduces *both* reward and frustration ratings (in line with hedonic blunting), the fMRI analysis exclusively focuses on reward. How does naloxone affect the neural processing of no-reward outcomes (and how do these trial types enter the GLM)? By extension, are the effects appropriately interpreted as decreases in the pleasure of rewards (e.g. Abstract) or do they also decrease the disappointment induced by not obtaining reward?

Additionally, the third paragraph of the Discussion seems to argue that the erotic stimuli but not the conditioned stimuli of the present task constitute rewards associated with incentive salience. By contrast, the fourth and sixth paragraph seems to argue that the conditioned stimuli carried motivational significance, which can be understood as a synonym of incentive salience, and yet that viewing of these stimuli was different from reward consumption. The potential for ensuing confusion should be minimized in the Discussion.

6) Data display: Figure 4 and the related supplementary figures show contrasts, of e.g., high versus low outcomes. To fully appreciate the effects of naloxone, the contrast estimates should also be split up, so that high, low and no outcomes are shown separately for placebo and naloxone in further supplementary figures. This is also relevant for reinforcing the discussion point that adaptive coding explains the relatively small differences between high and low rewards. In the presence of full adaptive coding, the difference of high vs. no reward should be similar to the difference of low vs. no reward and both differences should be significantly larger than zero. Is this the case?

7) Data reporting: For future reference and meta-analyses, tables with whole-brain findings at p<0.001 uncorrected would be helpful. Moreover, the authors state that they do not report SCR data for the naloxone condition as naloxone has been shown to reduce the inhibition of electrodermal activity by the bulbar reticular formation. Was the SCR elevated here? Could relative differences in SCR not be interpreted over and above a mean increase?

8) Materials and methods: The last paragraph of the subsection “Task”, describes the estimated monetary equivalence ranges of erotic pictures. However, as far as I could see the amounts actually used (and those won in total) in the task are not specified (also not in the first paragraph where one may expect them).

9) The second paragraph of the subsection “Task”, mentions maximum button press rate without specifying how it was determined.

10) A 10 mm radius seems rather large for a hypothalamus ROI. Can the authors independently confirm that the identified cluster indeed lies within the hypothalamus?

---

## [Author Response]

Major concerns:In the discussion, there were a number of concerns that were generally about the selective manner in which the results – particularly behavioral – were presented and analyzed. While the SCRs are presented in a relatively clear manner, they are analyzed by t-test. Reward ratings were presented and analyzed pairwise. And there was no distinction between outcome and anticipatory phase data. This then carried on into the manner in which the BOLD was analyzed. One might argue this was targeted, but the targets were not a priori and the justification was not clear. This selective focus made it hard to evaluate the task (reviewer 1, point 1) and caused some questions about whether other contrasts might have been more appropriate or shown different results (reviewer 1, point 1 and reviewer 3, point 2 and 6). It was felt that a fuller analysis and consideration of these questions was necessary. Relatedly, reviewer 2 felt that softening the strong distinction the authors are making between phases that distinguish incentive versus hedonic responses. This would perhaps go along with a fuller presentation and analysis of anticipatory vs outcome related behavior. Finally reviewer 3 had a variety of concerns regarding limitations that should be added to the manuscript – the impact of naloxone on mood, which presumably might underlie what they see and the low N for example should be mentioned.

We have addressed all these points in the paper and in addition provide detailed explanations of our changes when addressing individual points of all reviewers (see below).

In addition during the proof reading stage we found a little rounding error in the paper. We reported the effect in the left amygdala to be T(18)=2.83; p=0.006; p(corrected)=0.041. This was a rounding error, the corrected p-value before rounding is 0.0441 which must be rounded to 0.044.

Separate reviews (please respond to each point):

Reviewer #1:

In this study, the authors test in humans the hypothesis that pleasure during outcome presentation is selectively mediated by opioid mechanisms. Subjects performed a simple delay task in which they responded to cues that predicted the delivery of high versus low value erotic versus monetary rewards. After obtaining baseline measures, the same task was done following challenge with naloxone or placebo. They report that naloxone significantly affected ratings and SCR for the erotic but not the monetary outcomes, and this effect was associated with reduced activation in ventral striatum, orbitofrontal cortex, amygdala, hypothalamus and medial prefrontal cortex. The result is generally consistent with the idea that opioids mediate the pleasure of receiving reward distinct from anticipation and incentive salience and provides evidence of this in humans along with the potential circuit mediating the effects. Overall the experiments seem well designed and cleanly conducted and the results important.

We thank the reviewer for the positive assessment of our study as important.

However, while I found it interesting and potentially it seems like a novel and important study, I was a bit frustrated with how the results – particularly the behavioral results – were presented. Possibly my concerns are quite minor and may just involve a fuller presentation and analysis of the results. But if I follow the design correctly, it seems to me that the analysis should be a simple multifactor ANOVA comparing effects of placebo and drug on the different measures (SCR, ratings) over the different rewards – money and pictures. Yet instead the presentation of these data focuses in on specific comparisons and reports t-tests. I don't think these were a priori comparisons really, so this is not really appropriate. I think the authors need to present the results from each type of trial and analyze for the effects of the different factors.

We now provide the results of a repeated measures ANOVA with factors treatment and condition. This analysis revealed a significant main effect of condition, a trend for the main effect of treatment and importantly, a treatment by condition interaction. We have also included the order of treatment as a between-subject factor to test for residual order effects (see comment by reviewer #3). Neither the main effect of treatment order, nor the interaction with the other effects was significant.

However, we would like to stress that our study was motivated and designed shortly after a paper appeared which had tested the hypothesis of *decreased* pleasure and fMRI activation in the brain after application of naloxone (Petrovic et al., 2008). Indeed they observed that naloxone led to an *attenuation* of pleasure ratings for larger reward outcomes. Yet they did not observe any difference in the ventral striatum, as one could have expected based on rodent literature. We reckoned that the null result in the ventral striatum might be related to the little hedonic value of monetary rewards and were inspired by work of the Dreher lab (Sescousse et al., 2010) to combine an MID (Knutson et al., 2000) with erotic pictures to elicit more hedonic drive.

Accordingly, we hypothesized that naloxone would *decrease* pleasure ratings. Consequently we used paired t-tests (one-sided according to our hypothesis) for the different conditions and corrected for multiple comparisons.

This could be extended to showing SCR measures during the anticipatory versus outcome periods as well.

The SCR measures in Figure 2 show anticipation and outcome. However, as mentioned in the paper, we refrain from analyzing the SCR data with respect to the treatment effect, as it has been shown that SCR is directly affected by naloxone (Traore et al., 1998). This is supported by our raw SCR data which differed substantially between the naloxone and the placebo sessions (raw SCR values in arbitrary units of the analog-digital-converter were 502.2 ± 40.2 for placebo and 824.8 ± 107.3 for naloxone, averaged across all conditions).

This level of detail would make the results much easier to appreciate and more meaningful I think. As it is, I feel like I do not have a full picture of the behavior.The second issue I have is with the explanation for why the ratings and results were so different for the two types of rewards. The authors seem to minimize this (e.g. social desireability – what is this – embarrassment?), but I wonder whether this difference is important and potentially reflects the fact that different systems are mediating the response to one (erotic, primarily emotional and directly experienced) versus more cognitive and abstract to the other (money, the actual outcomes must be thought about). The authors allude to this at one point, but then do not say much about it. If this is a real difference, it should be addressed more, especially as many fMRI studies use money as the primary reward.

This is an interesting point. Absolute ratings show higher pleasure for money than erotic pictures, but only a significant decrease of pleasure by naloxone for erotic pictures. SCR for the placebo condition clearly show higher responses for erotic than for monetary outcomes. fMRI responses in all areas of interest also show higher responses for erotic outcomes. Based on the fact that two measures (SCR and fMRI) agree and the observation that behavioral ratings can be subject to social desirability (or embarrassment), we have argued that the absolute ratings between monetary and erotic rewards are difficult to compare, whereas social desirability would not affect the difference in ratings between naloxone and placebo treatments. However, we agree with this reviewer that we cannot rule out that different systems are mediating the response to one (erotic, primarily emotional and directly experienced) versus more cognitive and abstract to the other (money, the actual outcomes must be thought about). We have added this point to the Discussion.

Additional data files and statistical comments:My major concern is partly statistics – as outlined above.

Reviewer #2:

This manuscript by Buchel, Miedl, and Sprenger presents an interesting and compelling demonstration that hedonic reactions to pleasant outcomes are opioid mediated in people.The experiments seem well designed, the data are convincing, and the entire article is clear and scholarly. The authors deserve congratulations on a fine study. I have no major criticisms.

Thank you.

Minor point:While I agree with the authors' primary conclusion that they demonstrate an opioid mediation of outcome hedonic impact (the naloxone effects are especially compelling), and the authors also are perfectly right to divide anticipation vs. outcome temporal phases into psychological incentive salience vs. hedonic impact predominant categories. However, it should also be noted that the temporal anticipatory/consummatory divide is only a rough guide for motivation/hedonic categories, and temporal moments dos not always map on perfectly to the psychological categories.That is, even if anticipatory phase is primarily 'wanting', there can be some cue-elicited hedonic impact in the anticipatory phase (albeit low level) as a conditioned hedonic reaction as the authors note. There can also be an augmentation of incentive salience in the outcome phase, as in drug-priming of craving, or the 'you can't eat just one chip' or cocktail peanut priming effect of a food morsel. It is plausible that sexual incentive salience could also be heightened during outcome phase of sexual images. In my view, this consideration might justify slightly softening the authors assignment of psychological components to anticipatory/consummatory phases, but does not detract from their main conclusion.

We agree with this reviewer and have now slightly softened this point.

Similarly, regarding the authors' description in Discussion: "In a sequential conditioning task in rodents in which a first conditioned stimulus (CS1) was followed by a second (CS2) which was then followed by a sucrose reward, it could be demonstrated that hedonic effects were only observed for the outcome phase, but neither for CS1 nor CS2 (Smith et al., 2011)..….Furthermore, the effect of incentive salience at CS2 was dopamine and opioid dependent, whereas the hedonic effect linked to the outcome phase was only opioid dependent."I think in the Smith et al., 2011 study, there actually was also an opioid enhancement of conditioned anticipatory hedonic reactions to CS1 and CS2 (as well as a larger hedonic enhancement to sucrose outcome). The conditioned hedonic reactions were very low to begin with, as the authors note, but became slightly larger after NAc opioid stimulation. As the authors say, the UCS outcome hedonic enhancement was much greater, in keeping with their conclusions. Possibly the description could be slightly adjusted.

We have now adjusted this statement.

Reviewer #3:

Buchel et al. investigated the impacts of the opioid receptor antagonist naloxone on reward processing in healthy human volunteers using an adapted incentive delay task with sexual and monetary stimuli. Naloxone treatment was associated with a stronger reduction of subjectively reported pleasure while viewing erotic pictures compared to viewing symbols of monetary reward. Moreover, naloxone decreased the activation of reward-related areas primarily during "reward delivery".The manuscript is well-written and the research question is of high interest. The imaging methods are sound. However, the behavioral pharmacology part has some major flaws. Below you will find the several major concerns that have to be addressed.

We thank this reviewer for considering our research question of high interest. We regret that this reviewer considers our behavioral pharmacology as having major flaws. However, we hope that the points below change this view.

1) The authors did not assess mood/affective state ratings, which is a major shortcoming of the study.

We have to apologize for not including our mood and side effects ratings in the previous version of the paper. Mood and possible side effects were coarsely quantified using a 5-point Likert scale (not at all – very) with 12 items for mood and a 7-point Likert scale (not present – extreme) with 7 items for side effects. The mood rating scale included the following items: satisfied, rested, restless, bad, worn out, calm, tired, good, uneasy, cheerful, unwell, relaxed. The 7 item side effect scale included the following items: dry mouth, dry skin, blurred vision, lethargy, sickness, dizziness, headache. No significant differences between the naloxone and placebo session were observed. We therefore conclude that naloxone has no major side effects on mood at the dose used in our study.

The reviewer disagrees with the statement that "[naloxone] generally does not have noticeable effects on mood (Grevert and Goldstein, 1978; Petrovic et al., 2008; Kut et al., 2011)". There are a number of other studies clearly showing that naloxone infusions can induce aversive states such as dysphoria (Cohen et al., 1983, Arch Gen Psychiatry; Martin del Campo et al., 1994, Psychopharmacology; 2000, J Affect Disorder). Thus, the overall effect of mood on pleasure ratings remains unclear here. This has to be discussed as a strong limitation.

Apart from our mood and side effects ratings, which showed no difference between naloxone and placebo, we like to point out that we used an initial dose of 0.15mg/kg followed by a continuous infusion for a steady state (0.2mg/kg/h). Yet, the studies cited by this reviewer used higher bolus doses, e.g. the first study mentioned (Cohen et al., 1983) used doses between 0.3 and 4(!)mg/kg. The second study (Martin del Campo et al., 1994) used 0.2 mg/kg and 1 mg/kg naloxone. Even for the lower dose (which was higher than our bolus) they observed “no effect on overall mood (POMS),…” only effects in the afternoon after drug administration. The last study (Martín del Campo et al., 2000) again used a higher dose (0.2mg/kg) than we applied and half of the 14 participants investigated were depressed patients. Consequently, these studies are in good agreement with our observation that with a naloxone bolus of 0.15mg/kg no significant side effects and no effects on mood occur.

2) The reviewer is concerned regarding the choice of the sexual stimuli and the contrast build on the related picture categories (low vs. high erotic reward). Most likely, the supposed "low erotic reward" (women in swimsuits) category is already inducing strong activations of the reward system.

As can be seen in new Figure 4—figure supplements 5-12 (as requested by this reviewer), this is not the case. In all regions of interest, we observed higher activation for high as compared to low erotic reward. This is also supported by the pleasure ratings.

This interpretation is supported by the fact that women in swimsuits ("low erotic reward") induced much stronger SCR responses than high monetary rewards but that there was no difference compared to high erotic reward pictures.

We agree with this reviewer that skin conductance responses to erotic stimuli were significantly stronger as compared to monetary rewards, whereas SCR differences between high and low erotic stimuli were small. This could indeed indicate a ceiling effect in SCR responses for these stimuli. However, our other measures (behavioral VAS ratings and fMRI activation) actually show a consistent difference between low and high erotic pictures.

Interestingly a recent study has shown that explicit erotic pictures (comparable with "high erotic reward" stimuli used here) did not induce higher sexual arousal ratings in comparison to implicit erotic pictures (comparable with "low erotic reward" stimuli) (Schmid et al., 2015, Eur Neuropsychopharmacol). Thus, the question arises as to whether the chosen contrast is ideal for investigating the rewarding nature of sexual stimuli. More neutral pictures might have been a better choice. This has to be discussed as a major limitation.

We like to point out a few details of the study mentioned here (Schmid et al., 2015). The major difference is that a mixed sample of 15 male and 15 female volunteers was studied. The study employed an active condition in which volunteers could prolong the viewing time of stimuli by button presses. Surprisingly, the results show that volunteers prolonged viewing time of neutral stimuli much *more* than *any* erotic stimuli (both p<0.001). This could be indicative of a strong social desirability effect, where volunteers find it embarrassing to “work” for watching erotic stimuli. Nevertheless, this study also reported that “methylphenidate but not MDMA increased ratings of sexual excitation by visual stimuli with *explicit* sexual content …”. This modulation could not have been observed if only implicit stimuli were used. This is similar to our study, where the pharmacological effect was most pronounced for the high erotic condition. As can be seen in our new Figure 4—figure supplements 5-12 the effect of naloxone is evident for high erotic rewards, but not for low erotic rewards (and not for monetary stimuli). Only by including high erotic stimuli, we were able to reveal this important role of the opiate system, which we would have clearly missed had we only employed low erotic stimuli. In addition, we consider it important to use a parametric approach (i.e. comparing high to low) as opposed to comparing high erotic stimuli to baseline, as this is less specific and as this reviewer points out, can also be confounded by unspecific effects.

3) In light of the replication crisis in human neuroscience, the sample size is considered small (n=19) – a further limitation that has to be discussed. Related to that point, the authors report corrected and uncorrected p-values for the behavioral analyses but tested only one-tailed hypotheses. A more conservative approach would be appreciated, e.g., reporting and discussing only corrected results.

We agree with this reviewer and now mention this in the Discussion. However, we like to mention that previous studies have highlighted the advantage of longitudinal designs which is based on the fact that “each patient [is studied] as their own control” (e.g. (Martín del Campo et al., 2000) cited by this reviewer above).

4) Did the authors control for potential effects of order?

Yes potential order effects were reduced by randomizing the order of naloxone and placebo treatment.

Specifically, the very short test-retest interval (48h) suggests that order effects should be investigated, as naloxone may impact up- or down-regulation of opioid receptors far beyond 48h (Sirohi et al., 2007, J Pharmacol Exp Ther; Jenab and Inturrisi 1994, Mol Brain Res; Belcheva et al., 1992; Mol Pharmacol). Thus, the administration of naloxone on the first test day may have influenced results from the placebo condition on the second test day. The very short interval is another major weakness of the study.

We added a between subject factor “order” to the repeated measures ANOVA for the behavioral data (see above). This analysis did not reveal a significant main effect of order or an interaction with condition, treatment or a treatment by condition interaction. This suggests that randomization and the time between treatments was sufficient to avoid systematic carry over effects.

5) Interpretation and nature of effects: While the behavioral data suggest that naloxone reduces both reward and frustration ratings (in line with hedonic blunting), the fMRI analysis exclusively focuses on reward. How does naloxone affect the neural processing of no-reward outcomes (and how do these trial types enter the GLM)? By extension, are the effects appropriately interpreted as decreases in the pleasure of rewards (e.g. Abstract) or do they also decrease the disappointment induced by not obtaining reward?

This is a very important point. We now provide additional figures showing the effects of all conditions in each ROI for saline and naloxone (Figure 4—figure supplements 5-12). From these figures, it is clearly visible that the effect of naloxone in all regions is restricted to the reward (i.e. gain trials). In the miss trials most areas show no modulation by naloxone.

Additionally, the third paragraph of the Discussion seems to argue that the erotic stimuli but not the conditioned stimuli of the present task constitute rewards associated with incentive salience. By contrast, the fourth and sixth paragraph seems to argue that the conditioned stimuli carried motivational significance, which can be understood as a synonym of incentive salience, and yet that viewing of these stimuli was different from reward consumption. The potential for ensuing confusion should be minimized in the Discussion.

We thank this reviewer for pointing out this confusing paragraph. Rodent studies have converged on the observation that direct sexual behavior is decreased by opioids and increased by opioid antagonists. We have now put this directly in perspective with our findings and suggest that although the pleasure of viewing erotic pictures can be reduced by an opioid antagonist, this is distinct from direct sexual behavior which explicitly includes additional processes (e.g. motor output).

6) Data display: Figure 4 and the related supplementary figures show contrasts, of e.g., high versus low outcomes. To fully appreciate the effects of naloxone, the contrast estimates should also be split up, so that high, low and no outcomes are shown separately for placebo and naloxone in further supplementary figures. This is also relevant for reinforcing the discussion point that adaptive coding explains the relatively small differences between high and low rewards. In the presence of full adaptive coding, the difference of high vs. no reward should be similar to the difference of low vs. no reward and both differences should be significantly larger than zero. Is this the case?

We now provide the requested figures (Figure 4—figure supplements 5-12; one figure for each region of interest with 12 subplots for the different conditions and each subplot containing the data for the saline and for the naloxone run.

In particular, the figures for bilateral amygdala (Figure 4-figure supplement 9 and 10; row 2 column 3) and ventral striatum (Figure 4—figure supplement 5 and 6; row 2 column 3) highlight what we refer to as “adaptive”: The strongest responses is seen in the high erotic outcome condition for placebo treatment, and responses to other outcomes such as high monetary rewards (and low erotic and monetary rewards are blunted). We therefore argue that the system shifts its dynamic range to be able to represent the most rewarding stimuli (reward outcome high erotic), at the expense of other stimuli.

7) Data reporting: For future reference and meta-analyses, tables with whole-brain findings at p<0.001 uncorrected would be helpful.

We agree that meta-analyses can be of value. However, meta-analyses based on peak coordinates are prone to bias (e.g. coordinates in SPM are reported only with 8mm distance and effects with a p value slightly bigger than 0.001 might contribute significantly to an effect). We therefore, would rather encourage colleagues to exploit the contrast estimate images (which will be publicly available) for potential meta-analyses.

Moreover, the authors state that they do not report SCR data for the naloxone condition as naloxone has been shown to reduce the inhibition of electrodermal activity by the bulbar reticular formation. Was the SCR elevated here?

The raw SCR values (in arbitrary units of the analog-digital-converter and averaged across all conditions) were lower for placebo (502.2 ± 40.2) compared to naloxone (824.8 ± 107.3).

Could relative differences in SCR not be interpreted over and above a mean increase?

Absolute SCR values heavily depend on skin condition, environmental temperature and other factors. Therefore, the most sensitive way to analyze SCR within an experiment is to normalize (Z transform) the data. Unfortunately, in a longitudinal experiment as performed here, where SCR was measured on 2 days under naloxone and saline Z transforming each experiment separately potentially removes differences between treatments. Transforming the data across both treatments is problematic due to differences in electrode position, skin temperature etc. Furthermore, as mentioned in the manuscript SCR was only available for 14 volunteers.

8) Materials and methods: The last paragraph of the subsection “Task”, describes the estimated monetary equivalence ranges of erotic pictures. However, as far as I could see the amounts actually used (and those won in total) in the task are not specified (also not in the first paragraph where one may expect them).

The actual amounts used ranged from 0.1 – 5.6€ (0.85 ± 1.09€; mean ± sd) for the low amount and from 0.8 to 8.1€ (mean ± sd 2.71 ± 1.32€) for the high amount. This has been added to the manuscript. The valid response window for the MID was adjusted individually, so total gains cannot be meaningfully compared across volunteers or treatments.

9) The second paragraph of the subsection “Task”, mentions maximum button press rate without specifying how it was determined.

We estimated the maximum number of button presses for each individual before the equivalence experiment, by asking them to press a button as often as possible in a 5s period. This has been added to the manuscript.

10) A 10 mm radius seems rather large for a hypothalamus ROI. Can the authors independently confirm that the identified cluster indeed lies within the hypothalamus?

The radius of the hypothalamus ROI is larger as compared to the other ROIs, because it is centered close to the midline and thus covers the left and the right hypothalamus.